# Cognitive Deficits Following a Post-Myocardial Infarct in the Rat Are Blocked by the Serotonin-Norepinephrine Reuptake Inhibitor Desvenlafaxine

**DOI:** 10.3390/ijms19123748

**Published:** 2018-11-26

**Authors:** Mandy Malick, Kim Gilbert, Jonathan Brouillette, Roger Godbout, Guy Rousseau

**Affiliations:** 1Research Center, Hôpital du Sacré-Coeur de Montréal, Montréal, QC H4J 1C5, Canada; mandy.malick@usherbrooke.ca (M.M.); kim.gilbert3@gmail.com (K.G.); Jonathan.Brouillette@umontreal.ca (J.B.); Roger.Godbout@umontreal.ca (R.G.); 2Department of Pharmacology and Physiology, Université de Montréal, Montréal, QC H3C 3J7, Canada; 3Department of Psychiatry, Université de Montréal, Montréal, QC H3C 3J7, Canada

**Keywords:** desvenlafaxine, myocardial infarction, caspase-3, caspase-6, behavior, limbic system, memory

## Abstract

Myocardial infarction (MI) in animal models induces cognitive deficits as well as the activation of caspase in the limbic system; both can be blocked by 2 weeks of treatment following MI using tricyclic antidepressants or selective serotonin uptake blockers. Here we used three different treatment schedules to test the short- and long-term effects of the combined serotonin-norepinephrine reuptake inhibitor desvenlafaxine on post-MI-associated cognitive deficits and caspase activation. MI was induced in 39 young adult rats, and 39 rats served as sham-operated controls. Desvenlafaxine (3 mg/kg/day, i.p.) or saline was administered according to one of three schedules: (1) for 2 weeks, starting right after surgery; (2) for 16 weeks, starting 2 weeks after surgery; (3) for 16 weeks, starting right after surgery. Behavior was tested 2 weeks (social interaction, passive avoidance) and 16 weeks (forced swimming, Morris water maze) after surgery. Caspase-3 and caspase-6 activities were measured 16 weeks after surgery. At 2 and 16 weeks post-surgery, saline-treated MI rats displayed performance deficits compared to desvenlafaxine-treated rats, regardless of the treatment schedule. Caspase-3 activity was higher in the amygdala (medial and lateral) and hippocampal CA3 region in untreated MI rats, whereas caspase-6 activity was higher in the CA1 region. Caspase-6 activity correlated positively with deficits in the Morris water maze. These results indicate that, independently of treatment schedules, various treatment schedules with desvenlafaxine can prevent MI-associated cognitive deficits and decrease caspase activities in the limbic system.

## 1. Introduction

Cardiovascular disease is a risk factor for developing cognitive impairments [1] and the detailed mechanisms involved are being progressively identified. For example, apoptosis and activation of caspases were observed in the hippocampus (a brain structure involved in memory) and the amygdala (a brain structure involved in the regulation of emotions) 3 days after a myocardial infarct (MI), but not at 7-day post-MI [2,3,4]. This wave of apoptosis was shown to be followed at day 14 by cognitive deficits that are compatible with clinical signs of depression [5]. MI-induced apoptosis and cognitive deficits were prevented by early treatments with tricyclic antidepressants and selective serotonin uptake blockers, suggesting the contribution of early apoptosis in the limbic system to the later development of the post-MI behavioral syndrome [4,5]. One possibility is that neuronal loss due to post-MI apoptosis caused impaired trophic support or electrical stimulation of post-synaptic targets, leading to further downstream cell losses and ultimately to cognitive deficits [6]. Another possibility is that the activation of caspases, mainly caspase-8, may cleave different substrates such as caspase-3 and caspase-6 [7,8]. Caspase-6, originally identified as a key enzyme for apoptosis, has been shown to be involved in memory impairment when overactivated in the CA1 region of the hippocampus [9]. The caspase activation that accompanies the early phase of myocardial reperfusion may thus participate in the cognitive deficits that develop after MI. With that in mind, we hypothesized that preventing the early activation of caspases during the first two weeks of reperfusion should result in an attenuation of the post-MI early and delayed cognitive deficits. Treatment schedules with various onsets and offsets still need to be tested to further understand the pathophysiology of post-MI cognitive deficits.

Desvenlafaxine (DV) is a serotonin-norepinephrine reuptake inhibitor with proven antidepressant properties in humans [10]. Its use is not accompanied by significant adverse cardiovascular or cerebrovascular effects [11] so it could represent an interesting alternative for MI patients with depression. Since DV has been demonstrated to reduce apoptosis in the amygdala following MI in rats [12] it is reasonable to think that it may attenuate post-MI cognitive impairments.

The present study was designed to determine if: 1- DV could attenuate cognitive deficits after MI; 2- Early or delayed treatment with DV could be efficient in preventing post-MI cognitive deficits.

## 2. Results

### 2.1. Ischemic Damage

SS and area at risk (AR)/left ventricle (LV) percentages were similar between ischemia groups, with values around 30% and 65%, respectively (Table 1).

### 2.2. Social Interaction, 2 Weeks Post-MI

The two-way ANOVA discerned significant differences between groups (F_(1,53)_ 4.02; *p* = 0.05). Analyses of interactions indicated that the differences were due to MI-Veh rats which spent less time with congeners compared to Sham-Veh controls (F_(1,54)_ 11.95; *p* < 0.05). No difference was observed between Sham-DV and MI-DV (*n* = 10–18/group). There was a significant difference based on the MI factor (F_(1,53)_ 5.94; *p* < 0.05), with MI rats interacting less with their congeners than Sham rats. Untreated MI rats thus interact less with their congeners compared to the other groups and DV normalizes this impairment. See Figure 1.

### 2.3. Passive-Avoidance Step-Down Test, 2 Weeks Post-MI

The two-way ANOVA indicated no significant differences between groups (F_(1,46)_ 1.05; *p* > 0.05). However, main factors MI (F_(1,46)_ 6.93; *p* < 0.05) and DV treatment (F_(1,46)_ 6.38; *p* < 0.05) were significant. This was also reflected in the number of trials needed to learn the test, which was significantly different for the main factors (MI: F_(1,46)_ 11.38; *p* < 0.05 and DV treatment: F_(1,46)_ 4.32; *p* < 0.05) but not for interaction. Untreated MI rats thus need more trials to learn the task and DV reverses this. See Figure 2.

### 2.4. Forced Swimming Test, 4 Months Post-MI

ANOVA divulged significant between-group differences in immobility time (F_(5,56)_ 4.92; *p* < 0.05). *Post hoc* analysis revealed that MI-Veh rats were more immobile than Sham-Veh rats (*p* < 0.05). All other groups performed similarly to Sham-Veh rats. ANOVA also indicated significant between-group differences in swimming time (F_(5,56)_ 4.37; *p* < 0.05) and escape trial duration (F_(5,56)_ 3.65; *p* < 0.05). *Post hoc* tests revealed that Sham-DV and MI-DV rats swam more and tried less to escape than Sham-Veh rats (*p* < 0.05). The higher immobility time in untreated MI rats confirms the existence of behavioral despair while DV reverses this by increasing swimming time. See Figure 3.

### 2.5. Morris Water Maze (MWM) Test, 4 Months Post-MI

ANOVA indicated a difference between groups (F_(15,326)_ 2.097; *p* < 0.05). Further analysis revealed that only the MI-Veh curve was significantly different from Sham-Veh (F_(3,106)_ 4.822; *p* < 0.05). Similarly, the number of quadrants crossed by the animals during the test was significantly different between groups (F_(15,326)_ 2.042; *p* < 0.05). Further analysis showed that the performance of MI-Veh rats was impaired relative to that of the Sham-Veh rats (F_(3,106)_ 2.806; *p* < 0.05). No significant differences were observed in the probe test between groups: every groups spent about 3–4-fold more time in the target quadrant compared to the opposite quadrant. These data indicate that MI-Veh rats present a spatial memory deficit that can be reversed by DV. See Figure 4.

### 2.6. Caspase-3 and -6 Activities

Caspase-3 activity was significantly different between groups in the CA3 region, the lateral amygdala (LA) and medial amygdala (MA) 4 months after MI (CA3: (F_(5,34)_ = 4.47; *p* < 0.05); the MA (F_(5,26)_ = 5.65; *p* < 0.05); the LA (F_(5,30)_ = 6.30; *p* < 0.05) (Figure 5). *Post hoc* analysis indicated that MI-Veh rats showed significantly higher caspase-3 activity than Sham-Veh in all 3 regions. There were no differences between MI-DV and Sham-Veh rats, suggesting that DV reverses the increase in caspase-3 activity observed in MI-Veh rats.

In contrast, caspase-6 activity in the LA and MA regions was similar among all groups. In the CA1 region, however, caspase-6 activity was significantly different among groups (F_(5,38)_ = 3.56; *p* < 0.05). Further analysis pointed to differences between MI-Veh and Sham-Veh rats (*p* < 0.05). No other significant difference was apparent among the other groups (Figure 6).

Linear regression analysis indicated a significant positive correlation between caspase-6 activity in the CA1 region of the hippocampus and the time taken to find the platform in the Morris Water Maze (MWM) test on day 3 (*r*^2^ = 0.36; *p* < 0.05) (Figure 7).

## 3. Discussion

The present study indicates that MI leads to short and long-term cognitive deficits that can be attenuated by DV whether the treatment starts at the onset of reperfusion or 14 days after it. Results on the MWM test showed that the learning curve of MI rats (time and number of quadrants) was significantly different from those of the other groups. However, animals from the different groups performed similarly on the probe test and the visible platform test, indicating that swimming capacity *per se* was not affected.

The cognitive deficit ascertained in the present study extends those observed in 3-month-old rats tested 2 weeks post-MI [3,13,14]. In the latter groups of rats, deficits were already apparent in the Forced swimming test while performance in the MWM was comparable to those of control rats [3]. The effects of MI on the Forced swim test performance thus seem to last many months and an antidepressant such as DV appears to be efficient all along.

The picture differs in the case of the MWM test, where 3-month-old rats tested 2 weeks post-MI showed a normal performance while the untreated 8-month-old rats tested here displayed deficits that could be prevented by DV. The same phenomenon was observed with caspase-3 activation, which was found to be unaltered in different regions of the brain of 3-month-old rats tested 2 weeks post-MI while it increased in 8-month-old rats tested here, 4 months post-MI. This possibly reflects a long-term effect of MI on hippocampus-based performance such as spatial memory. In another pilot study [15] we have already shown a decreased performance on the MWM in 10-month old rats tested only 2 weeks after MI, suggesting that age rather than delay before testing is involved. 

The data obtained in the present study also indicates that caspase-3 activity increased in the CA3 region, where it was not possible to detect apoptosis during the first days of reperfusion [16]. This result is concordant with the hypothesis that contact with other neurons is important for cell survival. At the same time, increased caspase-3 activity was documented in the amygdala, one of the regions presenting apoptosis soon after reperfusion onset, and no caspase-3 activity was detected at 7-day post-MI in this region [12,16]. We have previously hypothesized that the activation of caspase-3 and caspase-8 during the first days of reperfusion can be related to the inflammatory state induced by the release of pro-inflammatory cytokines during myocardial infarction [13]. The long-term data in the present study supports the hypothesis of a second apoptosis wave at 4 months post-MI. Other studies have shown that the first wave of apoptosis is often followed by a second wave [17]. This second wave may be attributed to loss of neuronal activity or trophic support for other neurons [6,18].

Increased caspase-6 activity was observed in the CA1 region at 4 months post-MI. Caspase-6 is an enzyme widely expressed in the brain and, according to recent studies, its activation in the CA1 seems to be sufficient to cause memory deficits [9]. Knock-in mice expressing self-activated caspase-6 in the CA1 region develop age-dependent spatial and episodic memory impairment [9]. Caspase-6 has also been associated with different models of neurodegeneration, such as Alzheimer’s disease [19,20]. Here we found that higher level of Caspase-6 activity in the CA1 was significantly correlated with poorer performance on day 3 in the MWM test. Day 3 was chosen in the MWM test because it was near the inflection point of the learning curve.

There is a tight relation between caspase-3, -6 and -8 as proposed by different authors [21,22]. Activation of one of these caspases may lead to the activation of the others. Interestingly, DV treatment improved performance in the MWM test to a level similar to that in sham controls and reduced caspase-6 activity. It seems that short-term DV treatment is sufficient to avert the cognitive deficits seen at 4 months post-MI and suggests the importance of reperfusion onset as trigger. We have previously reported that DV attenuates apoptosis in the amygdala [12] and it has been reported that many antidepressants [23], such as DV, have some anti-inflammatory properties which may reduce the activation of the extrinsic apoptosis pathway.

Our data also confirm previous results that signs of behavioral impairment are observed 14 days post-MI [4,5,24] and extend the impaired period to 16 weeks after MI. Previous studies have documented that MI induces a behavioral syndrome that is akin to clinical signs of depression in humans [3,13,24,25]. This was confirmed in the present work with 2 of these tests at 2-week post-MI: untreated MI rats (MI-Veh) socialized less and were impaired in the passive-avoidance step-down test compared to the other groups. Testing 14 weeks later also showed that untreated MI rats (MI-Veh) still presented depression-like behavior based on the forced swimming test. 

The protocol implemented permitted the assessment of short- and long-term behavioral effects of MI and its treatment with DV. It demonstrated that DV treatment for 14–16 weeks post-MI alleviated behavioral depressive symptoms 4 months later compared to MI-Veh rats. Interestingly, DV exposure during the first 2 weeks post-MI was also efficient, showing that early DV administration has long-term outcomes in the present rat model. It is the second time that short-term treatment after reperfusion onset is found to be beneficial in preventing post-MI depression [26]. It challenges the necessity of long-term protocols to counter post-MI depression with antidepressants and suggests that inhibition or attenuation of signals, soon after reperfusion onset, could help in averting the development of post-MI depression. Whether this result is due to apoptosis reduction or attenuation of inflammatory responses [3,4,27,28,29] remains unanswered, and further investigation is undergone.

Our aim was to determine if MI could induce short- and long-term cognitive deficits and if DV could prevent them. However, the experimental design did not fully replicate the fact that, in humans, MI patients are discharged from the hospital with medications that were not used here, including beta-blockers and angiotensin-converting enzyme (ACE) inhibitors. Their own effects as well as potential drug-drug interactions were thus not tested here. Since inflammation could be involved in the development of post-MI cognitive deficits [13] the use beta-blockers [30] or ACE inhibitors [31] would be particularly relevant since they have anti-inflammatory properties. 

Another limitation is that although ischemic damage seemed to be similar between MI groups, heart function may have differed. One study has found that venlafaxine increased heart function [32] while another did not [33]. Further research should therefore evaluate heart function using ejection fraction, which seems to be more precise than dP/dt to estimate heart function [34].

## 4. Materials and Methods

### 4.1. Experimental Groups

Animal care and handling procedures were approved by the local Animal Care Committee (Grou28; February 2014) and followed Canadian Council for Animal Care guidelines.

A total of 78 young, adult, male Sprague-Dawley rats (4 months old at the start of the protocol) were purchased from Charles River (St-Constant, Québec, Canada). They were housed individually under constant temperature (22 °C) and humidity (40–50%), with food and water available ad libitum. The light period was 12 h long and started at 8:00 a.m.

The rats were randomly assigned to MI (*n* = 39) or sham (*n* = 39) groups. In MI rats, the left anterior coronary artery was occluded for 40 min (see details of the surgical procedure below). Sham-operated rats were submitted to the same protocol except that the coronary artery was not occluded. 

After surgery, MI and sham rats were randomized into 1 of 2 treatment subgroups receiving either DV or saline (vehicle). In MI rats, treatment was delivered according to 1 of 3 possible schedules: (1) restricted to the first 2 weeks post-MI; (2) started 2-week post-MI and continued up to 16 weeks post-MI; (3) lasted 16-week post-MI (Figure 8). Sham controls received treatment according to schedule 3 only. Two subgroups, 1 sham and 1 MI, were given a vehicle for 16 weeks. The final number of rats per subgroup is reported in the Results section. The following abbreviations denote the various subgroups tested: Sham-Veh = Sham controls treated with saline; Sham-DV = Sham controls treated with DV; MI-Veh = MI rats treated with saline; MI-DV = MI rats treated with DV; MI-Veh-DV = MI rats treated with saline for 2 weeks and then with DV; MI-DV-Veh = MI rats treated with DV for 2 weeks and then with saline. DV was administered for the first 3 days i.p. (3 mg/kg, diluted in 0.5 mL of saline) every morning and was diluted in drinking water on the following days. Water quantity ingested was calculated every morning. We opted to change the way of administration for two reasons: (1) animals during the first days after MI could have some difficulties to drink; (2) we also wanted to reduce the manipulation of the animals to avoid a bias in the behavioral tests.

### 4.2. Surgical Procedure

Anesthesia was induced with ketamine/xylazine (50 mg/kg and 5 mg/kg i.m., respectively) and maintained with 1.5% isoflurane ventilation. Electrocardiogram and heart rate were monitored during surgery. After left thoracotomy, the left coronary artery was occluded with 4-0 silk suture and plastic snare. Ischemia was confirmed by ECG changes and the presence of epicardial cyanosis. The ligature was removed after 40 min of occlusion to initiate reperfusion. The first DV injection was administered 5 min after the onset of reperfusion. The thorax was closed, and the rats were returned to their cages. Antibiotic (15,000 IU penicillin G s.c.) and analgesic (2 mg/kg, 0.2 mL buprenorphine s.c.) were administered. The animals received a second buprenorphine dose the next morning. 

### 4.3. Scar Sections (SS), Area at Risk (AR) Measurement and Brain Dissection

Four months after surgery, the animals were restrained in a cone bag and rapidly sacrificed by decapitation. Their brains and hearts were removed swiftly and placed in a dish kept on crushed ice. The hearts were washed in saline, and the left anterior coronary artery was occluded at the same site as during MI surgery and infused through the aorta with 2 mL of 0.5% Evans Blue. They were then frozen at −80 °C, and 4 transverse slices were cut (2 mm). SS were clearly visible and expressed as AR percentage (SS/AR × 100), to estimate the degree of damage due to ischemia. The AR was also expressed as percentage of LV area (AR/LV × 100). The LA and MA and the hippocampus (CA1 and CA3 regions) were isolated rapidly, snap frozen in liquid nitrogen, and kept at −80 °C until needed.

### 4.4. Behavioral Tests

Rats were tested in the morning. Behavioral tests at 2 weeks were used to demonstrate the presence of behavioral signs compatible with our post-MI rat model of depression [13]. Two other behavioral tests were administered 4 months after surgery to document the presence of long-term cognitive deficits.

#### 4.4.1. Social Interaction Test

This test was conducted on day 13 post-surgery. Two rats, regardless of their experimental condition, were placed together in a clean cage for 10 min. During this period, 2 observers (blinded to the experimental conditions) each monitored 1 of the 2 animals. Interactions were scored when 1 rat smelled, touched the nose, or groomed the other rat. The duration of interaction with the other rat was quantified.

#### 4.4.2. Passive-Avoidance Step-Down Test

This test was performed on day 14 post-surgery. The rats were placed individually on a Plexiglas platform (14 cm × 19 cm) in a test chamber (14 cm × 33 cm). An electrifiable grid (14 cm × 14 cm) stretched alongside and 2.5 cm lower than the Plexiglas platform. To be successful on this test, the animals needed to stay on the top platform for 60 s over 3 consecutive trials. If they stepped down onto the lower platform, resting all 4 paws on the grid, they would receive a mild shock to the feet (1 s, 8 mA) and were then returned to their cages for 30 s. The number of trials needed to meet the test criteria and the time needed to learn the test were noted. 

#### 4.4.3. Forced Swimming Test

Four months post-surgery, rats were placed individually in a transparent 25-cm diameter pool filled to a depth of 30 cm with 22–25 °C water, with no possible escape route. Two observers without knowledge of the experimental condition used identical chronometers to measure the time the rats were immobile, swimming or trying to escape from the pool. The test was conducted over a 2-day period: day 1 comprised 15 min of habituation, and day 2 entailed the actual 5-min test. 

#### 4.4.4. MWM Test

At 4-month post-surgery, rats were placed in a pool (150-cm diameter, 50 cm depth) filled to 25 cm with water maintained at 22–25 °C and made opaque with powdered milk. A submerged platform was positioned just below the surface of the water. The rats were tested on 6 trials every day, 5 min apart, for 6 consecutive days. They underwent probe testing on day 7. The platform was removed, and time spent in the quadrant where the platform used to be was calculated along with time spent in the opposite quadrant. On day 7, the platform remained visible. The number of quadrants crossed and the time taken to reach the platform were noted.

### 4.5. Caspase-3 and Caspase-6 Activity

Biochemical measures were taken 4 months after surgery. Cytosolic proteins were extracted in lysis buffer (1% Triton X-100, 0.32 M Sucrose, 10 mM Tris (pH 8.0), 5 mM EDTA, 2 mM DTT, 1 mM PMSF, 10 µg/mL Leupeptin, 10 µg/mL Pepstatin A, 10 µg/mL Aprotinin). Enzymatic reactions were produced in reaction buffer (50 mM Tris (pH 7.5), 5 mM MgCl_2_, 1 mM EGTA, 0.1% CHAPS, 1 mM DTT) with 25 µg of proteins and fluorogenic substrate: Ac-DEVD-AMC (40 µM) for caspase-3, and Ac-VEID-AMC (40 µM) for caspase-6. Reactions were incubated at 37 °C for 180 min for caspase-3 and 90 min for caspase-6 and stopped with the addition of 0.4 M NaOH and 0.4 M glycine buffer. Fluorescence was quantified with a spectrofluorometer (Photon Technology International, Lawrenceville, NJ, USA) at excitation wavelength of 365 nm (caspase-3) or 325 nm (caspase-6) and emission wavelength of 465 nm (caspase-3) or 435 nm (caspase-6).

### 4.6. Statistical Analysis

The data are reported as mean (± standard error of the mean). Statistical analyses were undertaken with 2-way ANOVA on the social interaction test and the passive-avoidance step-down test with MI and DV as main factors. Biochemical tests and the forced swimming test were analyzed by one-way ANOVAs, followed by Dunnett *post hoc* tests when significant. Sham-Veh served as controls. These analyses were conducted with SPSS, version 19 (Armonk, NY, USA). The analysis of the MWM results was performed by normalizing the data with reference to the performance of the Sham-Veh group on day 1 set as 100%. Then, the data were fitted with the 1-phase exponential decay equation provided by Prism version 7.0c (La Jolla, CA, USA). Group curves were compared to Sham-Veh curves to detect significant differences. Numbers of quadrants crossed by the animals were analyzed similarly. Differences in the data obtained on the probe test and the visible platform test were tested by ANOVA. Linear regression coefficients between caspase-6 activity in the CA1 region of the hippocampus and MWM performance on day 3 were calculated by Prism version 6.0e (La Jolla, CA, USA). Day 3 was selected because it was near the inflection point of the learning curve. *p* ≤ 0.05 was considered to be significant in every statistical test.

## 5. Conclusions

MI induced cognitive deficits that may be prevented by DV as well as depressive-like behavior. A short-term treatment, starting at the onset of the reperfusion, can attenuate these deficits.

## Figures and Tables

**Figure 1 ijms-19-03748-f001:**
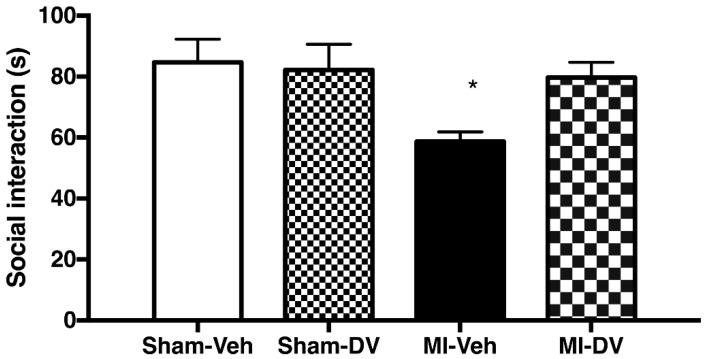
Social interaction. MI rats treated with the vehicle interacted less with other rats than Sham rats treated with the vehicle. MI rats treated with DV performed at the same level as sham rats treated with the vehicle or with DV. *n* = 10–18/group. * *p* < 0.05. MI: myocardial infarction; Veh: vehicle; DV: Desvenlafaxine.

**Figure 2 ijms-19-03748-f002:**
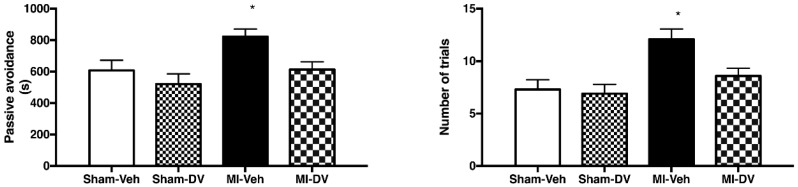
Passive-avoidance step-down test. Left panel: Time to success refers to the time spent by the rats on the platform they should not escape from. Right panel: Number of trials required before learning the task. The performance of MI rats treated with the vehicle was impaired compared to the other groups of rats. MI rats treated with DV performed at the same level as sham rats treated with the vehicle or with DV. *n* = 9–16 per group. * *p* < 0.05. MI: myocardial infarction; Veh: vehicle; DV: Desvenlafaxine.

**Figure 3 ijms-19-03748-f003:**
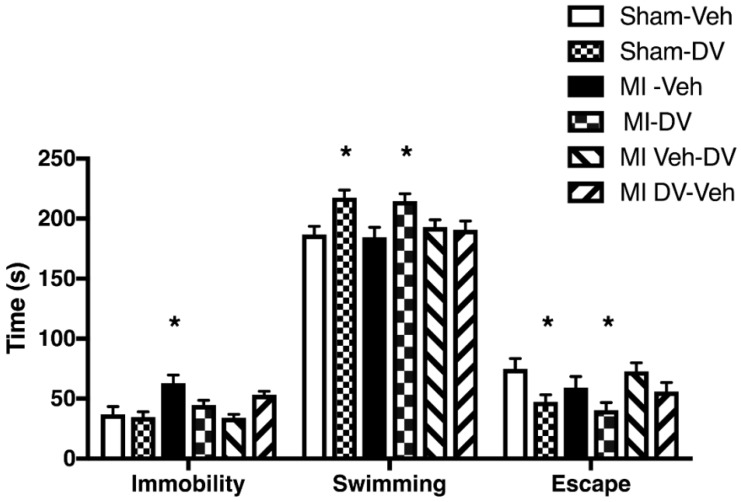
Forced swimming test. MI rats treated with the vehicle were more immobile than the Sham rats treated with the vehicle and they swam less than the Sham rats treated with the vehicle or with DV. MI rats treated with DV performed at the same level as sham rats treated with the vehicle or with DV on measures of immobility and swimming. Escape trial duration was significantly longer in MI rats treated with the vehicle than the DV-treated Sham rats and MI rats. *n* = 9–12 per group. * *p* < 0.05. MI: myocardial infarction; Veh: vehicle; DV: Desvenlafaxine.

**Figure 4 ijms-19-03748-f004:**
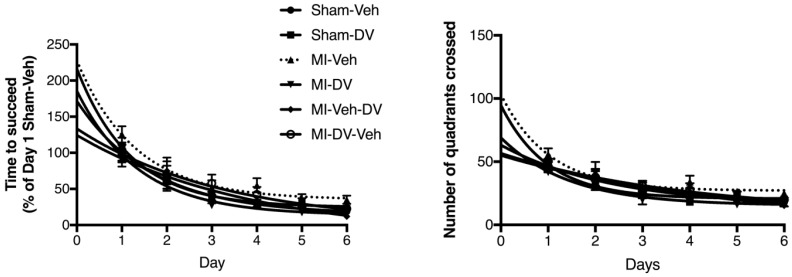
Morris Water Maze test. Left panel: Time needed to reach the target platform expressed as a percentage of time needed by sham rats treated with the vehicle. Right panel: Total number of quadrants crossed (6 trials per day) before finding the target platform. The dotted lines with triangle markers represent the results obtained by vehicle-treated MI rats. When tested 4 months after surgery, vehicle-treated MI rats took more time to find the platform and crossed more quadrants to find the platform compared to any of the other groups; *n* = 9–12 per group. (*p* < 0.05). The performance of the MI rats treated with DV was the same as that of the vehicle-treated sham rats. MI: myocardial infarction; Veh: vehicle; DV: Desvenlafaxine.

**Figure 5 ijms-19-03748-f005:**
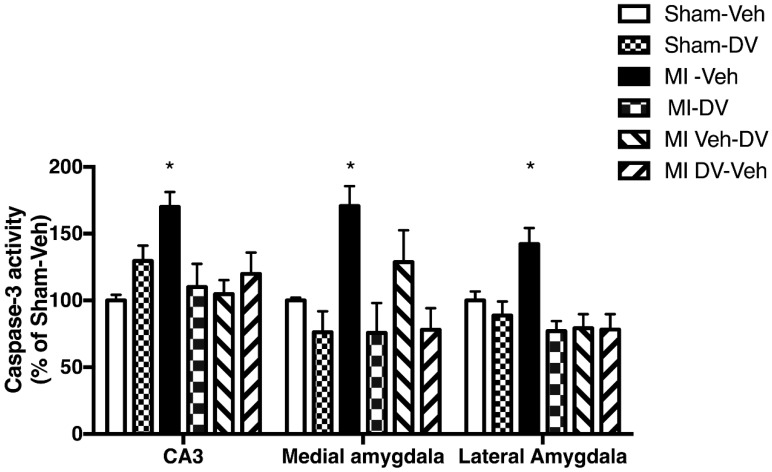
Caspase-3 activity. Casapse-3 activity in the CA3 region of the hippocampus, the medial amygdala, and the lateral amygdala of MI-Veh compared to Sham-Veh rats 4 months after surgery. Caspase-3 activity is higher in the vehicle-treated rats compared to any other group; caspase-3 activity is not different between DV-treated rats and vehicle-treated sham rats. *n* = 6–8 per group. * *p* < 0.05.

**Figure 6 ijms-19-03748-f006:**
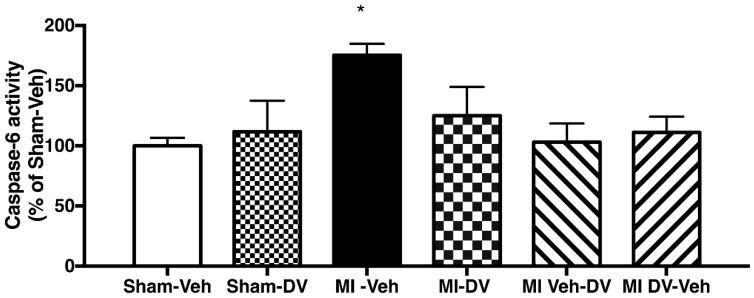
Caspase-6 activity in the CA1 region of the hippocampus 4 months after surgery. Caspase-6 activity in the CA1 region of the hippocampus is higher in vehicle-treated MI rats compared to any other group; caspase-6 activity is not different between DV-treated rats and vehicle-treated sham rats.; *n* = 6–8 per group (* *p* < 0.05).

**Figure 7 ijms-19-03748-f007:**
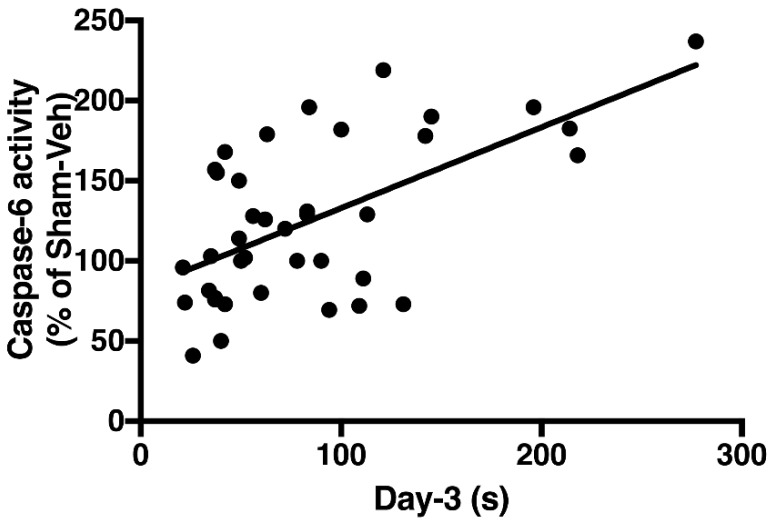
Correlation between spatial memory performance and caspase-6 activity in the CA1 region of the hippocampus 4 months after surgery. Data are taken from day 3 With all groups of rats pooled together, the linear regression test showed a significant positive correlation between the time needed to reach the target platform (on day 3 of a 6 days series—see Figure 4). *r*^2^ = 0.36; *p* < 0.05.

**Figure 8 ijms-19-03748-f008:**
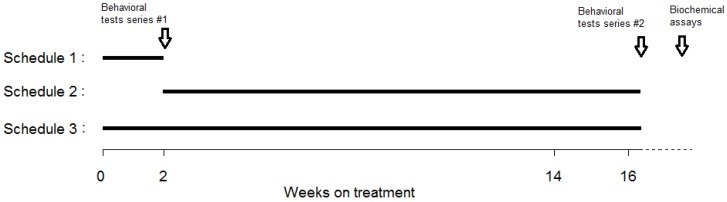
Protocol timeline. Schematic representation of the treatment protocol. MI rats treated with DV were distributed in schedules 1, 2 and 3. Lines represent DV treatment. Sham rats treated with DV were distributed only in schedule 3. Other MI and sham rats received only vehicle. Behavioral test series #1: social interaction test and passive-avoidance step-down test; behavioral test series #2: forced swimming and Morris Water Maze (MWM) tests. “Water tests” were grouped at the end of the experiment.

**Table 1 ijms-19-03748-t001:** Scar section and area at risk (AR).

Group	Scar Section/AR	AR/LV
MI-Veh	31.3 ± 1.6	66.4 ± 1.5
MI-DV	29.0 ± 1.9	63.8 ± 1.5
MI-Veh-DV	31.1± 1.6	65.5 ± 2.3
MI-DV-Veh	26.3 ± 1.1	64.7 ± 2.0

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
