# Peer review of "Cognitive Deficits Following a Post-Myocardial Infarct in the Rat Are Blocked by the Serotonin-Norepinephrine Reuptake Inhibitor Desvenlafaxine"

_ijms, 2018, doi:10.3390/ijms19123748_

Round 1

Reviewer 1 Report

In the study, titled "Cognitive deficits following a post-myocardial infarct in the rat are blocked by the serotonin-norepinephrine reuptake inhibitor desvenlafaxine", Malick M et al, describe the role of desvenlafaxine (DV) in preventing cognitive defects in rats after myocardial infarction injury. Studies have shown that MI in patients is followed by depression and apoptosis in hippocampal regions of the brain suggesting a link between myocardial injury and cognitive disorders. In this respect, the idea of the manuscript is interesting and previously overlooked. Nevertheless, there are significant concerns with experimental plan and interpretation of the data that must be addressed as highlighted below;

Does treatment with DV reduce circulatory serotonin/norepinephrine levels or locally in the brain tissue? Moreover, can the authors provide catecholamine levels before and after treatment with DV under MI conditions.

Its not clear what group of MI rats and treated with DV were used for the various cognitive tests. In the methods, authors describe that DV was administered under 3 different schedules. Did the authors see any difference with timing and length of DV administration?

The main premise of the study is to document the beneficial role of DV administration on cognitive defects in patients with MI. However, MI patients are typically prescribed a number of drugs to alleviate adverse cardiac effects and increase cardiac contractility which is intended to slow the progression of disease but at the same time leads to improvement in symptoms. Therefore, the model that authors use is not really depictive of the clinical condition for 2 reasons

Animals used have MI and a progressively worsening cardiac function since there is no administration of drugs that would halt the adverse effects.

The drug-drug interactions of DV with other cardioprotective medications is not taken into account.

Overall, results section lacks details making comprehension of the results difficult.

Author Response

Reviewer 1

Comments: Does treatment with DV reduce circulatory serotonin/norepinephrine levels or locally in the brain tissue? Moreover, can the authors provide catecholamine levels before and after treatment with DV under MI conditions.

Answer: Unfortunately these data are not available. It would have been necessary to put the rats in a restrainer to be able to draw blood from the awakened animals, which is a stressful condition that would have influenced the catecholamine levels. The same would have occurred under isoflurane anesthesia. It was therefore not possible to perform these measurements.

Comments: It is not clear what group of MI rats and treated with DV were used for the various cognitive tests. In the methods, authors describe that DV was administered under 3 different schedules. Did the authors see any difference with timing and length of DV administration?

Answer: No difference between the different schedules of DV was observed for the various cognitive tests at 4 months. We clarified this issue in the summary.

Comments: The main premise of the study is to document the beneficial role of DV administration on cognitive defects in patients with MI. However, MI patients are typically prescribed a number of drugs to alleviate adverse cardiac effects and increase cardiac contractility which is intended to slow the progression of disease but at the same time leads to improvement in symptoms. Therefore, the model that authors use is not really depictive of the clinical condition for 2 reasons

Animals used have MI and a progressively worsening cardiac function since there is no administration of drugs that would halt the adverse effects.

The drug-drug interactions of DV with other cardioprotective medications is not taken into account.

Answer: These arguments are interesting and we added a new paragraph in the discussion, even though they do not disqualify our methods. Even if it is true that we did not use all the medications that humans usually receive, our study was designed to determine if cognitive deficits could be developed after MI and if DV could prevent it. Interestingly, in humans, depression is observed after MI independently of the drugs administered after MI and data is accumulating that MI is a risk factor to develop cognitive deficits. Our study indicates that cognitive deficits could, at least, be retarded after MI in presence of DV. Concerning the second part of the comment, the drug-drug interactions was indeed not taken into account but, once again, this is not directly relevant to the purpose of the article. Our purpose is to determine if cognitive deficits following MI could be prevented with DV.

Meanwhile, these points are interesting and we add a new paragraph in the discussion.

Reviewer 2 Report

The aim of the authors is to investigate the heart-brain connection, specifically the functional (cognitive) consequences of the post-MI apoptotic waves in the amygdala and hippocampal CA3 region. The authors report that the antidepressant desvenlafaxine can prevent MI-associated cognitive deficits and decrease caspase activity in the limbic system.

The authors are to be praised for focusing on a really novel and revolutionary field, ie the herat-brain connection. Specifically, they are focusing on MI-associated cognitive deficits, a field that remains understudied but with huge clinical relevance. The methods are solid, the results are robust, the conclusions are supported by the data, and the manuscript reads well.

However, this peer reviewer raises the following concerns:

Major issues:

The main result is the reduction in apoptosis with treatment with desvenlafaxine. However, in science, all important results should be confirmed by two independent techniques. Therefore, the authors should perform an additional assessment of apoptosis in the amygdala, eg. TUNEL or (if histological sections are not available, Bax/Bcl).

The message about MI (heart disease) affecting the brain is novel. But the authors should hypothesize why an ischemic insult in the heart will cause apoptosis in a distant organ like the brain, eg sympathetic overdrive post-MI?, systemic inflammation post-MI (as demonstrated in Nature. 2012 Jul 19;487(7407):325-9) also affecting the brain? LV dysfunction impairing blood flow to the brain?

The authors should also mention that antidepressant (particularly affecting serotonin, such as desvenlafaxine) may have an effect reducing platelet hyper-reactivity (as shown in Am J Cardiol. 2002 Feb 1;89(3):331-3), which could also be beneficial for the patient

Do the authors have data about LV systolic function? Was EF or dP/dt worse in the MI+vehicle group? Please also mention that LVEF is a superior assessment of LV systolic function than dP/dt as demonstrated in Am J Physiol Heart Circ Physiol. 2012 Apr 1;302(7):H1423-8

Minor issues:

Fig 5: The authors present only two lines instead of 5 because they combine 4 groups into 1 single line. They should not do that, they should present the 5 group lines independently

Author Response

Reviewer 2

Comments: The main result is the reduction in apoptosis with treatment with desvenlafaxine. However, in science, all important results should be confirmed by two independent techniques. Therefore, the authors should perform an additional assessment of apoptosis in the amygdala, eg. TUNEL or (if histological sections are not available, Bax/Bcl).

Answer: The high number of measurements performed in this study required that we keep to a minimum measurements that have been repeatedly performed in the past. Indeed, we have already published results indicating that DV reduces the activity of caspase-3 as well as TUNEL positive cells in the same models, 3 days post-MI (Kaloustian et al. 2008), as mentioned in the present manuscript. Our previous results have shown that apoptosis peaks 3 days post-MI and apoptosis is no longer active 2 weeks post MI (Wann et al. 2007). We focused our efforts on the activity of the different caspases since their sensitivity seems to be higher.

Comments: The message about MI (heart disease) affecting the brain is novel. But the authors should hypothesize why an ischemic insult in the heart will cause apoptosis in a distant organ like the brain, eg sympathetic overdrive post-MI?, systemic inflammation post-MI (as demonstrated in Nature. 2012 Jul 19;487(7407):325-9) also affecting the brain? LV dysfunction impairing blood flow to the brain?

Answer: According to our previous work (Bah et al. 2010; Gilbert et al. 2013), we believe that the inflammation post-MI is indeed a key component for the results observed in the brain. We have previously reported that the inhibition of inflammation with different treatments resulted in a diminution of apoptosis and/or reduction of cognitive deficits (Gilbert et al. 2014, 2015).

Comments:  The authors should also mention that antidepressant (particularly affecting serotonin, such as desvenlafaxine) may have an effect reducing platelet hyper-reactivity (as shown in Am J Cardiol. 2002 Feb 1;89(3):331-3), which could also be beneficial for the patient

Answer: This is a stimulating point. In fact our protocol is modelling the reverse situation of Shimbo et al 2002: we trigger signs and symptoms of depression in naïve rats by inducing an acute coronary event (myocardial infarction). It would indeed be interesting to assess platelet hyper-reactivity in our model, before and after treatment with antidepressants.

Comments: Do the authors have data about LV systolic function? Was EF or dP/dt worse in the MI+vehicle group? Please also mention that LVEF is a superior assessment of LV systolic function than dP/dt as demonstrated in Am J Physiol Heart Circ Physiol. 2012 Apr 1;302(7):H1423-8

Answer: Unfortunately we do not have LV systolic function data and this is an important point that should be addressed in our next study. We included this issue in the limitation section of the discussion.

Minor issues:

Fig 5: The authors present only two lines instead of 5 because they combine 4 groups into 1 single line. They should not do that, they should present the 5 group lines independently

Answer: The individual curves are now presented for each group.

Round 2

Reviewer 1 Report

All reviewer concerns have been addressed